# A Review of the Diagnosis and Management of Premalignant Pancreatic Cystic Lesions

**DOI:** 10.3390/jcm10061284

**Published:** 2021-03-19

**Authors:** Margaret G. Keane, Elham Afghani

**Affiliations:** Department of Medicine, Johns Hopkins University, Baltimore, MD 21287, USA; mkeane13@jhmi.edu

**Keywords:** pancreatic cystic lesions, pancreatic cancer, intraductal papillary mucinous neoplasm, mucinous cystic neoplasm, endoscopic ultrasonography, magnetic resonance imaging, computer tomography, diagnosis, management, surveillance

## Abstract

Pancreatic cystic lesions are an increasingly common clinical finding. They represent a heterogeneous group of lesions that include two of the three known precursors of pancreatic cancer, intraductal papillary mucinous neoplasms (IPMN) and mucinous cystic neoplasms (MCN). Given that approximately 8% of pancreatic cancers arise from these lesions, careful surveillance and timely surgery offers an opportunity for early curative resection in a disease with a dismal prognosis. This review summarizes the current evidence and guidelines for the diagnosis and management of IPMN/MCN. Current pre-operative diagnostic tests in pancreatic cysts are imperfect and a proportion of patients continue to undergo unnecessary surgical resection annually. Balancing cancer prevention while preventing surgical overtreatment, continues to be challenging when managing pancreatic cysts. Cyst fluid molecular markers, such as *KRAS*, *GNAS*, *VHL*, *PIK3CA*, *SMAD4* and *TP53*, as well as emerging endoscopic technologies such as needle-based confocal laser endomicroscopy and through the needle microbiopsy forceps demonstrate improved diagnostic accuracy. Differences in management and areas of uncertainty between the guidelines are also discussed, including indications for surgery, surveillance protocols and if and when surveillance can be discontinued.

## 1. Introduction

Globally, pancreatic cancer is the twelfth most common cancer but the seventh most common cause of cancer-related death. In 2018, there were an estimated 459,600 new cases and 432,000 deaths from the disease [1]. The incidence in the Western population is increasing, with the highest being in Europe and North America [1,2,3,4]. It is estimated to become the second leading cause of cancer-related death by 2030 [5]. A study of 3.9 million cancer patients globally found pancreatic cancer to have the lowest five-year survival rates, ranging from 7.9% in the United Kingdom to 14.6% in Australia [6]. Due to the lack of overt symptoms in earlier stages of the disease, most patients are diagnosed at a stage when curative resection is no longer possible, leading to the low survival rate [7]. Patients diagnosed at an early stage have a substantially better prognosis and survival compared to those diagnosed with more advanced stages, as more patients diagnosed in earlier stages are likely to be candidates for surgical resection with improved survival [8]. Even if the tumor is not amendable to surgical resection, a lower tumor burden results in less chemoresistance, therefore making chemoradiotherapy treatments more effective [9,10]. Therefore, early diagnosis in pancreatic cancer has become a recognized healthcare priority [7,11].

Pancreatic cystic lesions (PCL) are an increasingly common incidental finding. They are present in 1.2–2.6% of patients undergoing abdominal computed tomography (CT) [12,13] and in up to 13.5% of patients undergoing abdominal magnetic resonance imaging (MRI) for non-pancreatic indications [14]. The incidence increases further with age, with approximately 10% of individuals over 70 years old undergoing a CT being found to have PCL [15]. PCL have a broad differential diagnosis [16]. Table 1 shows the characteristics of the common PCL. Intraductal papillary mucinous neoplasms (IPMNs) and mucinous cystic neoplasms (MCNs) are of particular importance because these are considered precursor lesions to pancreatic cancer [17,18,19,20,21,22]. In contrast to the other precursors, such as pancreatic intraepithelial neoplasia (PanIN) which can only be identified on surgical histopathology, IPMNs and MCNs can be easily identified on cross-sectional imaging [23,24,25,26,27] Given that approximately 8% of all pancreatic cancers are believed to arise from these lesions, this offers an opportunity for early cancer detection [28].

### 1.1. Classification of IPMNs and MCNs

An IPMN is a mucin producing tumor that arises from the pancreatic duct. They are equally common in men and women. There are three types of IPMNs, which are differentiated based on morphologic differences. Main duct IPMNs (MD-IPMN) are characterized by involvement with the main pancreatic duct (MPD), and identified by a dilated MPD (≥5 mm) without an associated cyst or other cause for ductal obstruction. Branch duct IPMNs (BD-IPMN) arise from a branch off the MPD, and are identified as unilocular or multilocular pancreatic cyst with communication with the MPD, which measures <5 mm. Mixed-type IPMNs (MT-IPMN) meet criteria for both MD and BD IPMNs (Table 1). Furthermore, IPMNs can be histologically classified as gastric, intestinal, pancreaticobiliary or oncocytic based on cellular morphology and mucin (MUC) gene expression and tissue architecture [33]. Studies have suggested that knowing the epithelial subtypes may be of prognostic importance (Table 2) [34]. 

Gastric-type IPMNs have the best prognosis, as they are typically small BD-IPMNs with low-grade dysplasia (LGD), and have a 5-year survival of >90%. Prognosis following resection is good, with 5- and 10-year survival rates of over 90%. Intestinal-type IPMNs are often involving the MPD and are MT or MD-IPMNs with high-grade dysplasia (HGD). Prognosis of intestinal IPMNs are less favorable, with 5- and 10-year survival rates of 70% and 50%, respectively, when associated with pancreatic cancer. Pancreatobiliary-type IPMNs arise from BD, MT or MD-IPMNs but are exclusively high-grade neoplasms, and seen up to 80% of cases associated with invasive pancreatic cancer. Five- and 10-year survival rates are 50% and 0%, respectively. Oncocytic IPMNs are rare but tend to occur in younger patients. They arise in MD-IPMNs with HGD and around 50% are associated with invasive cancer. Patients with oncocytic-type IPMNs with associated cancer have a 5- and 10-year survival of 60% and 40%, respectively [35].

MCNs, on the other hand, are lined by tall columnar mucin producing epithelial cells and in contrast to IPMNs, are surrounded by ovarian-type stroma [16,36]. There is predominance for these lesions to be detected in middle-aged women [37] (Table 1). 

### 1.2. IPMN/ MCN Progression to Invasive Cancer

The natural history and longitudinal risk of malignancy in IPMNs and MCNs are poorly understood. Although these lesions can progress from low-grade to high-grade dysplasia and ultimately pancreatic cancer, not all IPMNs or MCNs progress to cancer within a patient’s lifetime. Each type of IPMN is associated with different rates of malignant transformation. In surgically resected BD-IPMN, the risk of malignant transformation has been reported to be between 6 and 51%. MD and MT- IPMNs are recognized to have higher rates of malignant transformation, ranging between 35–100% [23,24,25,26]. The risk of malignant transformation in MCNs have been reported to be between 0–34% [29]. Regardless, the data on the natural history of IPMNs and MCNs have limitations. Natural history studies that rely on surgical specimens include a disproportionate number of high-risk lesions so may overestimate the true cancer risk whereas cohort studies without histologic proven IPMNs and MCNs [38] may underestimate cancer risk.

Pancreatic cancer in IPMN can arise directly from the PCL (an associated cancer) or from the pancreatic parenchyma away from the IPMN (concomitant cancer), which occurs in between 9–44% of cases [39,40]. IPMN-associated and concomitant cancers have a better prognosis than non-IPMN pancreatic cancers. A recent systematic review revealed an improved 5-year survival for IPMN cancers (OR 0.23, 95% CI 0.09–0.56). Median survival ranged from 21 to 58 months in the IPMN cancers compared to 12–23 months in the non-IPMN related cancer group. It was noted that IPMN cancers were frequently found as stage 1 disease (OR 4.40, 95% CI 2.71–7.15) so it is possible that the improved survival is actually due to earlier detection [41]. 

Whole exome and targeted sequencing of small cohorts of IPMNs and MCNs have identified genetic alterations in oncogenes and tumor suppressor genes, which drive progression to dysplasia and ultimately cancer. Like in pancreatic cancer, one of the earliest genetic alterations in IPMNs are thought to be in KRAS and GNAS [42,43,44]. Over time, mutations in tumor suppressor genes such as RNF43, CDKN2A, TP53 and SMAD4 occur which drives the progression to invasive cancer [43,44]. A targeted analysis of larger cohorts has confirmed that these gene mutations correlate with the grade of dysplasia and histological subtype [45,46]. However, targeted next generation sequencing of IPMNs has suggested there is considerable intratumoral genetic heterogeneity in these lesions and several different molecular alterations are present in different parts of the cyst [47]. It is likely that this combination of genetic alterations drive the transition from a noninvasive precursor lesion to invasive cancer in IPMNs [48,49]. 

MCNs are lined by columnar mucinous epithelium [16] and like IPMNs, are also now classified pathologically into a three-tiered system with associated LGD, HGD or pancreatic cancer [50]. Like IPMNs, MCNs also harbor genetic changes that lead to tumor progression and ultimately the development of invasive cancer. KRAS mutations are found in 3–100% of MCNs [42,43,51]. The frequency of KRAS alterations also seems to increase with grade of dysplasia [51,52]. GNAS mutations are commonly found in IPMNs but are not found in MCNs; however alterations in RNF43 have been found in 12% of low-grade MCNs and 25% of high-grade MCNs [42]. Loss of CDKN2A/p16 may also play a role in progression to cancer in IPMNs as it is a common finding in MCNs with HGD but is absent in MCNs with LGD. Similarly TP53 is present in 25–56% of MCNs with HGD or cancer, but not in MCNs with LGD [51]. Similar to IPMNs, loss of SMAD4 expression appears predominantly in MCNs with invasive cancer. In one study, which examined 36 MCNs, SMAD4 expression was retained MCNs with LGD or HGD but was lost in 86% of MCNs with invasive cancer [53]. 

The timeline for progression from IPMN/MCN with LGD to invasive cancer remains poorly understood. Mathematical modelling of the carcinogenesis of PanINs, suggests the progression from PanIN 1 to pancreatic cancer could take up to 35 years, of which 12 years includes the progression from PanIN-3 to pancreatic cancer [54]. Studies classify PCLs as low-risk or high-risk, with high risk having characteristics of high-risk stigmata, defined as presence of obstructive jaundice, enhancing mural nodule ≥5 mm, and main pancreatic duct ≥10 mm or worrisome features, defined as presence of pancreatitis, cyst ≥3 cm, enhancing mural nodule <5 mm, thickened cyst wall, main pancreatic duct of 5–9 mm, abrupt change in caliber of pancreatic duct, lymphadenopathy, increased serum level of carbohydrate antigen 19-9, and a cyst growth rate ≥5 mm/2 years [27]. A systematic review of retrospective surveillance cohorts found that low-risk IPMNs defined as BD-IPMNs without mural nodules, had an approximate 8% chance of progressing to invasive cancer within 10 years while BD-IPMN with worrisome features had 25% chance of progressing to cancer in 10 years [55]. A genetic analysis of the evolutionary timeline of the malignant transformation of IPMNs suggests a window of approximately 3 years to progress from HGD to invasive cancer [56]. These studies suggest progression to cancer occurs over at least several years in IPMNs/MCNs, which supports the utility of surveillance programs that enable the early detection of pancreatic cancer and high-risk lesions. 

### 1.3. Guidelines for the Management of IPMN and MCN

There are currently five major guidelines on the management of IPMN and MCN: the revised International Consensus guidelines [30], the European evidence based guidelines on the management of pancreatic cystic neoplasms [57], the American Gastroenterology Association (AGA) guidelines on the management of asymptomatic PCL [58], the American College of Gastroenterology (ACG) clinical guideline on the diagnosis and management of PCL [59] and the American College of Radiology (ACR) white paper on the management of incidental pancreatic cysts [60]. The similarities and differences between the recommendations are discussed below. The quality of the evidence on which management recommendations are based in IPMN and MCN is often low, so many of the guidelines are formed from expert and consensus opinion (Table 3 and Table 4).

## 2. Diagnosis of IPMN and MCN

A definitive diagnosis of the type of PCL is made through histopathology. However, most patients with low-risk PCL will not undergo surgical resection. Attempting to determine cyst type is important to determine risk of malignant transformation and optimal management. Therefore, clinical and radiological features are used to characterize and diagnose a suspected IPMN or MCN.

### 2.1. Predicting Malignant Transformation in IPMN

#### 2.1.1. Symptoms and Risk Factors

Most IPMNs and MCNs are detected incidentally when abdominal imaging is performed for another indication [61,62]. On direct questioning, up to half of patients may report mild or vague symptoms such as abdominal pain or bloating. However, it is unlikely that these symptoms are directly related to the PCL. The presence of a number of clinical symptoms has been associated with malignant transformation in patients with IPMNs/MCNs. Jaundice in patients with IPMNs/ MCNs may occur as a result of compression of the common bile duct as the cyst enlarges in the head of the pancreas. Jaundice can also potentially occur due to mucin plugging the ampulla and distal bile duct or rarely, as a result of direct tumor invasion of the bile duct. Significant weight loss is also suggestive of malignant transformation [62,63,64].

New-onset diabetes is a recognized early symptom of pancreatic cancer, and therefore, the relationship between diabetes and IPMN has been explored. In a systematic review of diabetes in patients with IPMNs, which included 27 studies with histologically confirmed IPMNs, diabetes was associated with an increased risk of harbouring MPD involvement (risk ratio 1.43, 95% CI: 1.21–1.69, *p* < 0.001), HGD (risk ratio 1.27, 95% CI: 1.01–1.59, *p* = 0.04) or pancreatic cancer (risk ratio 1.61, 95% CI: 1.33–1.95, *p* < 0.001) [65]. In another study, using a prospectively maintained cohort of 442 patients with a suspected mucinous cyst without worrisome features or high-risk stigmata, the development of new-onset diabetes mellitus was strongly associated with progression to high-risk stigmata (HR = 11.6; 95% CI, 3.5–57.7%), but was not associated with the development of worrisome features after controlling for other risk factors. This is of importance in patients with larger cysts (2–3 cm), as they were found to have a 5-year adjusted cumulative risk of progression to developing high-risk stigmata of 53.5% (95% CI, 19.6%–89.9%) compared to only 7.5% (95% CI, 1.6%–15.2%) in patients without new-onset diabetes [66].

Another important symptom that can suggest a malignant IPMN is acute pancreatitis. In a study of 488 patients with IPMNs, followed in a tertiary referral center in Korea, acute pancreatitis or acute recurrent pancreatitis occurred in 7%. Acute pancreatitis occurred more frequently in MD or MT-IPMN as compared to BD-IPMN (14% vs. 5%, respectively (*p* = 0.002)). In the 24 patients with IPMN-associated cancer who underwent surgical resection, 8% were found to have HGD and 4% had invasive cancer [67]. In another large surgical cohort of 325 patients with an IPMNs, acute pancreatitis was reported in 21% prior to surgery. Some reported a single episode, while others had up to 10 distinct attacks. On multivariate analysis, acute pancreatitis was an independent predictor of intestinal subtype (OR 4.69, 95% CI 2.48–8.84, *p* < 0.001), malignancy (OR 1.97, 95% CI 1.07–3.63, *p* = 0.029), and MD IPMN (OR 1.87, 95% CI 1.02–3.43, *p* = 0.044). Patients with acute pancreatitis due to an IPMN should be managed as if harboring a malignant IPMN likely involving the MPD [68].

#### 2.1.2. Tumour Markers

Serum tumor markers alone cannot be used to diagnose IPMN or MCN or reliably differentiate them from other PCL [69], but some studies have found certain biomarkers to be predictive of malignant transformation. The most widely studied biomarker is the carbohydrate antigen 19-9 (CA 19-9) which is elevated in approximately 85% of patients with pancreatic cancer [70]. In a recent meta-analysis of fifteen studies with 1629 patients, an elevated serum CA 19-9 had a sensitivity and specificity of 52 and 88% for detecting an IPMN with pancreatic cancer [71]. In HGD, CA19-9 is elevated much less frequently than in patients with invasive cancer (47.9% vs. 11.4%, *p* < 0.001) [72]. Although 37 units/mL is the standard cut-off for CA19-9, a cut-off of 100 units/ml had the highest accuracy for detecting invasive carcinoma (93% vs. 83%) [72,73]. Elevated serum CA 19-9 has therefore been included as a worrisome feature in some of the revised clinical guidelines [30,57], but as it has limited utility in detecting HGD, which is the optimal screening target for patients with IPMN/MCN, it is still not widely performed in most surveillance patients, unless there is a suspicion of invasive cancer. It is also important to note that an elevated CA 19-9 can be seen in benign conditions, such as inflammatory bowel disease, acute and chronic pancreatitis, cirrhosis, cholangitis, benign biliary obstructions, ovarian cysts, heart failure, hashimoto’s thyroiditis, diverticulitis and rheumatoid arthritis [74]. 

Other serum biomarkers that have been evaluated include the neutrophil-to-lymphocyte ratio (NLR). In a series of 272 patients with a resected IPMN, a NLR of greater than 4 was significantly associated with pancreatic cancer, independent of MPD size, presence of mural nodules, jaundice or cyst size. Similar to CA19-9, a higher NLR was not associated with HGD but was associated with invasive carcinoma, which again may affect its utility as a screening tool in IPMN [75]. Carcinoembryonic antigen (CEA) is another tumor marker that has been found to be elevated in patients with pancreatic cancer [76]. A study comparing the role of CA 19-9 and CEA found that while serum CA 19-9 and/or CEA were elevated in 80% of cohort with invasive IPMN, CA 19-9 performed better than CEA [73]. 

Ultimately, a series of clinical signs and a panel of biomarkers may provide the best diagnostic accuracy for detecting HGD or invasive cancer in those with IPMNs and MCNs. In a recent study using integrated data modelling of multiple compounds in plasma and cyst fluid, mucinous and serous cysts were found to have significant difference in lipid pathway alterations. Mucinous cysts were discriminated from and serous cysts with a 100% accuracy and HGD and invasive cancer was detected with a 90.06% accuracy [77]. Further studies are required to validate these promising findings.

#### 2.1.3. Imaging

The current work-up of newly diagnosed PCL consists of gadolinium-enhanced MRI with magnetic resonance cholangiopancreatography (MRCP) or a pancreatic protocol CT. On cross sectional imaging, a PCL is classified as a “suspected” IPMN or MCN based on morphology. A MD-IPMN is suspected when there is an abrupt dilation of the MPD. A BD-IPMN is suspected when there is dilation of the side branches into a ‘grape- like’ cystic lesion that connects to MPD. A MT- IPMN has features of both a BD and a MD-IPMN. Most IPMNs occur in the head of the pancreas (70%) [61], with up to half of patients with a BD-IPMN having multifocal disease [78] (Table 1). 

In a recent meta-analysis that includes 70 studies with a total of 2297 patients who had undergone resection of an IPMN with a mural nodule, the presence of an enhancing mural nodule had a positive predictive value of 62% for advanced neoplasia on final pathology. The study also found the size of the enhancing mural nodule had a substantial effect on predicting advanced neoplasia, but no reliable size cut-off could be identified as only a few case series had measured mural nodules prior to resection [79]. The IAP and European guidelines have therefore incorporated enhancing mural nodules as an indication for surgery [30,57]. The cut off of 5 mm however, is arbitrary and requires further validation.

In most of guidelines, surgical resection is recommenced for patients with a MPD of 10 mm or larger, with close surveillance often with EUS when the duct is between 5 and 9 mm (Table 3). A retrospective single study of patients who underwent surgical resection of an IPMN found MPD dilation to be the best predictor for advanced neoplasia. In the final cohort of 1688 patients, of those that underwent resection, a MPD dilation of more than 10 mm was the only independent predictor of invasive cancer (OR 6.34) [79]. In a retrospective multicenter study of 901 patients, MPD dilatation between 5 mm to 9.9 mm was associated with increased odds ratio of HGD (OR = 2.74; 95% CI = 1.80–4.16) and invasive cancer (OR = 4.42; 95% CI = 2.55–7.66). The trend was even more overt when MPD dilatation was more than 10 mm (OR = 6.57 for HGD and an OR = 15.07 for invasive cancer). A MPD cutoff of 5 to 7 mm was therefore found to discriminate between malignant and benign lesions [80]. In older patients (over 65 years old), even minimal MPD dilation of 3–5 mm appeared to be important and predictive of high-risk lesions. This is not part of current guideline recommendations, but in a study of 923 patients who underwent surgical resection for an IPMN, without history of pancreatitis or jaundice, MPD <5 mm, cyst size <3 cm, no mural nodules, negative cyst fluid cytology for adenocarcinoma, or serum carbohydrate antigen 19–9 (CA 19–9) <37 U/L, minimal MPD dilation (OR 11.3, 95% CI 2.40–53.65; *p* = 0.002) was associated with high-risk of malignant transformation [81]. Abrupt change in the caliber of the pancreatic duct is also suggestive of malignancy in BD or MT IPMN. A recent meta-analysis of 40 articles, including 6301 patients with pathologically proven IPMN found an abrupt change in the caliber of the MPD was strongly predictive of HGD or pancreatic cancer (OR 7.41, CI 2.49–22.06) [82]. It has therefore been included as a worrisome feature in the International Consensus guidelines [30]. Another more common cause of a dilated pancreatic duct is chronic pancreatitis and differentiating the two can be challenging. Typically chronic pancreatitis is associated with risk factors such as smoking and excess alcohol consumption. In addition to MPD dilation on imaging there is also parenchymal or ductal calcification. A recent small study of patients undergoing pancreatoscopy found pancreatic duct stones can also be present in MD IPMN or chronic pancreatitis [83]. In this scenario, ERCP with per oral pancreatoscopy can be used to visualize the pancreatic duct and aid differentiation between these two pathologies.

MCNs typically arise in the body or tail of the pancreas and on cross sectional imaging are mostly unilocular macrocystic lesions. They can cause partial pancreatic ductal obstruction when they become large in size. Peripheral calcifications may form an “egg shell. The imaging features of MCNs which are predictive of malignant transformation include: Male sex (OR 3.72; 95% CI, 1.21–11.44; *p* = 0.02), located in pancreatic head and neck (OR 3.93; 95% CI, 1.43–10.81; *p* = 0.01), larger size (OR 1.17; 95% CI, 1.08–1.27; *p* < 0.001), presence of a solid component or mural nodule (OR 4.54; 95% CI, 1.95–10.57; *p* < 0.001), and MPD dilation >5 mm (OR, 4.17; 95% CI, 1.63–10.64; *p* = 0.003) [84].

#### 2.1.4. Metabolic Imaging

There are a number of studies that have explored the utility of F-18-fluorodeoxyglucose positron emission tomography (18-FDG-PET) in detecting malignant IPMN. A recent systematic review that included 10 articles and 419 patients found 18-FDG-PET was more sensitive, specific, and accurate than the current high-risk stigmata, as defined in the International Consensus guidelines for detecting malignant IPMNs (80%, 95%, and 87% vs. 67%, 58%, and 63%, respectively) [85]. However, the overall number of patients in these studies is small and there can be variations in PET/CT scanners and image interpretation, which can lead to false positive results. Although PET-CT can be considered as an adjunct to current diagnostic tools, particularly in lesions that are being considered for surgical resection, further studies are required to define its role in diagnostic algorithms [86]. Current guidelines make limited recommendations on the role of 18-FDG-PET in the management of IPMNs.

### 2.2. When to Perform EUS

Differentiation of non-malignant, premalignant and malignant cysts is important, as management is different. EUS is often undertaken when high-risk stigmata or worrisome features are present and where its findings will change management, e.g., indications for surgical resection. An EUS examination can determine the cyst size, location, wall thickness, presence of focal wall irregularity, associated mass or mural nodule, septae, echogenic debris or mucus and dilation of the MPD. When a fine needle aspiration (FNA) is performed, cyst fluid can be immediately assessed for the “string-sign” which suggests a mucinous lesion. In a study of 98 histologically proven cases, the sensitivity, specificity, positive predictive value, and negative predictive value of the string sign for diagnosis of mucinous cysts were 58%, 95%, 94% and 60%, respectively [87].

Pancreatic cyst fluid can also be analyzed for CEA, amylase, cytology and molecular markers (e.g., KRAS and GNAS). EUS morphology has a sensitivity of between 56% to 78%, and a specificity of 45% to 67% for differentiating an IPMN or MCN from other types of cysts [88,89]. Cytology has a sensitivity of 28% to 73% but a specificity of 83% to 100% for identifying mucinous cysts from other cysts [90]. The sensitivity is often low in PCL as many samples are paucicellular [91]. Some groups have therefore performed EUS with a fine needle biopsy needle and targeted the cyst wall to improve cellularity. With this approach, diagnostic adequacy of the cytology sample obtained increased to 65%, and in lesions with a solid component or with a malignancy it increased to 94.4% and 100%, respectively [92] Cyst fluid CEA is commonly used to differentiate mucinous from nonmucinous cysts, and has a higher accuracy for detecting mucinous cysts than either EUS morphology or cytology [88]. However, the cut-off level for CEA has been debated. Studies have shown that a cut off >800 ng/mL has a sensitivity of 38% but a specificity of 98% [93]. The optimal level that is used clinically is >192 ng/mL, which has a reported sensitivity of 73% and specificity of 84% [93]. A low CEA (<5 ng/mL) has a specificity of 95% and sensitivity of 50% for non-mucinous cysts (such as a pseudocyst or serous cystadenoma). However, it does not predict malignant transformation [88,89]. The presence of a string sign and CEA concentration ≥200 ng/mL when combined has a diagnostic accuracy of 89% [87]. A low glucose level in pancreatic fluid (≤41 mg/dL) has also been found to be predictive of a mucinous cyst. An advantage of this test over CEA is that it requires minimal fluid and it can provide an immediate result as it can be measured with a glucometer. A recent systematic review that included 31 studies with 5268 patients, found glucose performed better than CEA for mucinous cysts diagnosis with sensitivities of 91% (95% CI, 0.86–0.94) and 67% (95% CI, 0.65–0.70), specificities of 75% (95% CI, 0.68–0.82) and 80% (95% CI, 0.76–0.83), and areas under the ROC curve of 0.95 and 0.79, respectively [94]. A high amylase is found in cyst fluid from multiple types of cyst including pseudocysts, IPMNs, MCNs and serous cysts, so has limited diagnostic utility. Low amylase levels are rare in pseudocysts and are therefore helpful in only excluding this diagnosis [90].

Different types of PCL have different and specific genetic mutations that are detectable in small amounts of pancreatic cyst fluid which can be used to help determine the cyst subtype [95]. A mutation in *KRAS* occurs in IPMNs or MCNs, while a mutation in GNAS is found almost exclusively in IPMNs. The presence of mutation in VHL, with no other mutations, has 100% specificity for a serous cystadenoma [96,97]. Combining clinical features with molecular markers can improve the diagnostic accuracy of EUS further. In a recent retrospective multicenter study of 860 individuals referred for surgery for a PCL, the combined molecular and clinical panel was more accurate than current clinical features alone, and use of these markers would have decreased the number of unnecessary operations by 60% [96,97]. Molecular markers have also shown utility in identifying IPMNs and MCNs that harbor HGD or early invasive cancer. From a genetic perspective, growing knowledge suggests that early lesions are quite heterogeneous, but as they progress to HGD, they have a smaller number of homogeneous genetic drivers [96,97]. Early preliminary studies have shown that the presence of *PIK3CA*, *SMAD4* and *TP53* in cyst fluid are promising markers of malignant transformation, identifying almost 80% of IPMNs with HGD or cancer [98]. 

The revised International Consensus guidelines recommends that an EUS is performed in all suspected IPMNs with “worrisome features” or in surveillance alternating with MRI once cysts are >3 cm [30]. The European consensus guidelines also recommends performing EUS as an adjunct to other imaging modalities only when the results of EUS-FNA are expected to change the clinical management [57] (Table 4). However, an EUS does have some limitations as well as risks. First, the EUS is an invasive procedure with a risk of perforation, infection and bleeding (0.6%) [99]. In addition, EUS requires the use of anesthesia. Additionally, obtaining sufficient fluid for cytological and biochemical assessment in smaller PCL (<2 cm) can be challenging, especially in mucinous cysts when the contents are viscous and difficult to aspirate [91]. Some groups therefore debate the utility of EUS in surgical decision-making [100], but there has been a growing interest in novel diagnostics that can enhance the utility of EUS in PCL. 

### 2.3. Novel and Emerging EUS Guided Diagnostic Approaches

In patients with high-risk PCL, pre-diagnostic tests are known to be imperfect and a proportion of patients continue to undergo unnecessary pancreatic resection every year. A number of advancements in endoscopy including contrast-enhanced EUS, confocal laser endomicroscopy and through the needle biopsy forceps have emerged as enhanced diagnostic techniques in diagnosing PCL.

#### 2.3.1. Contrast Enhanced EUS

Contrast-enhanced harmonic endoscopic ultrasonography (CH-EUS) is a technology that combines an intravenous contrast agent-based with harmonic imaging by EUS. The two most commonly used contrast agents are SonoVue™ (Bracco, Milan, Italy) and Sonazoid™ (GE Healthcare, Oslo, Norway). Distinguishing between benign and malignant pancreatic masses, especially when small, by EUS alone, is a recognized clinical challenge. CH-EUS can be used to help differentiate these from malignant pancreatic tumors by demonstrating a hypoenhancing pattern compared with normal tissue, which is isoenhancing. Contrast- enhanced EUS can also aid discrimination between mural nodules and mucin clots in PCL. A meta-analysis of 70 studies with 2297 undergoing surgical resection of IPMN reported a CH-EUS had a PPV of 62% for detecting advanced neoplasia on final pathology [101]. CH-EUS is associated with intraoperator variability and although contrast agents are promising adjuncts to EUS at present, they cannot replace cytological tissue sampling for diagnostic purposes. 

#### 2.3.2. Confocal Laser Endomicroscopy

Needle based confocal laser endomicroscopy (nCLE) can provide real-time optical histology of the cyst wall during EUS-FNA. Using a laser scanning unit and the AQ-Flex miniprobe (Cellvizio; Mauna Kea Technologies, Paris, France), which can be passed through a 19G FNA needle, real time imaging of the epithelial lining of the cyst wall can be obtained. Initial studies to data have found EUS-nCLE to be a safe adjunct to routine EUS-FNA and have established diagnostic criteria for many of the common PCL [102]. nCLE has high specificity (>80 %) but sensitivity has varied by cyst type (69–95% for SCN, 59–95% for IPMN and 67–95% for MCN) [103,104,105,106,107]. Subsequent studies have also shown that there is good inter- and intraobserver agreement [108].

As outlined above, pathologically IPMN can be classified subtype (gastric, intestinal, pancreatobiliary, or oncocytic) and level of dysplasia (low or high grade dysplasia or presence of invasive cancer). Although this information is prognostically beneficial, it is usually only available after surgical resection. There has therefore been an interest in determining this information pre-operatively with nCLE, which can provide pathology in real time. In a pilot study of four patients with different subtypes of IPMNs, nCLE of the oncocytic subtype showed a unique appearance of papillae that were thicker and demonstrated a fine honeycomb pattern [105]. Another small study looking at the feasibility of nCLE in differentiating levels of dysplasia in IPMNs found increased papillary epithelial “width” and “darkness” to have a sensitivity and specificity of 90% and 91%, respectively, for detecting HGD or invasive cancer [109]. These initial findings require further validation.

#### 2.3.3. Through-the-Needle Biopsy

A through-the-needle microbiopsy forceps (Moray™, US Endoscopy, Mentor, OH, USA) can be passed through a 19G FNA needle. The forceps are opened within the cyst and are used to obtain larger tissue fragments from the cyst wall. A prospective study on 114 patients showed through-the-needle biopsy had a diagnostic yield of 83% compared to 38% with cytology [110]. A systematic review of microbiopsy forceps in PCL that included 9 studies with 454 patients found a diagnostic yield of 69.5% (95%CI 59.2–79.7), for the through-the-needle microbiopsy forceps vs. 28.7% (95% CI 15.7–41.6) for cytology. Sensitivity and specificity for the through-the-needle microbiopsy forceps in mucinous pancreatic cysts was 89% and 95%, respectively. Adverse events (intracystic bleeding and pancreatitis) occurred in 8.6% (95% CI 4.0–13.1) of patients [111]. Groups have also explored combining the techniques of nCLE and microforceps biopsy during the same EUS procedure. The diagnostic yield for cytology, microforceps and nCLE was 34.1%, 75.0% and 84.1%, respectively. When cytology, microforceps and nCLE were combined, the diagnostic yield increased to 93.2% and led to a change in management in 52.3% of cases [112].

#### 2.3.4. Deep Learning and Artificial Intelligence

The application of deep learning and artificial intelligence (AI) in diagnosing PCL has been explored in a number of recent publications. In one study of 85 patients, AI using deep learning was used to construct a diagnostic algorithm. CEA, CA19-9, carbohydrate antigen 125, and amylase in the cyst fluid, as well as sex, cyst location, connection of the MPD, cyst subtype and cytology were used to form the algorithm. The area under receiver-operating curve for the diagnostic ability in malignant cysts was 0.719 for CEA, 0.739 for cytology and 0.966 for AI using this algorithm. The AI algorithm had a sensitivity, specificity, and accuracy of 95.7%, 91.9%, and 92.9%, respectively which was significantly higher than CEA or cytology [113]. Pilot studies have also shown AI has a role in improving the interpretation of novel diagnostics such as nCLE [114]. These promising findings will require further validation.

## 3. Management of IPMN and MCN

### 3.1. Surveillance

Surveillance of IPMN and MCN provides the opportunity for early detection and potentially surgical curative surgery. Surveillance should therefore be offered to patients as long as they remain surgically fit enough and willing to undergo surgical resection [30,57,58,59,60]. However, differentiating IPMNs and MCNs correctly from all other PCL pre-operatively is a recognized clinical challenge, and as a result, a high number of patients are entering long-term surveillance annually [115,116].

The best modality for surveillance of IPMNs and MCNs has not been established, and therefore guidelines vary in their recommendations. For most patients, MRI is preferred method for surveillance as it avoids repeated exposure to ionizing radiation and provides improved delineation of the pancreatic duct and presence of an enhancing mural nodule or internal septations. However, there are ongoing concerns about possible gadolinium deposition in the brain, kidney and bone after repeated use of certain contrast agents in patients with normal renal function [117]. Some patients find MRI scans claustrophobic and they take considerably longer to perform than a CT scan, which only takes a few minutes. EUS or a pancreas protocol CT can therefore be considered as the primary surveillance tools in patients who cannot have or choose not to have MRI with MRCP [30,57,58,59,60].

For IPMNs without high-risk or worrisome features, the cyst size guides the frequency of surveillance in the International Consensus guidelines. In multifocal IPMNs, surveillance intervals are based on the size of the largest IPMN [30]. Size alone correlates imperfectly with malignancy in IPMN as cancers have occasionally been observed in small IPMN (<2 cm) with other worrisome features. The AGA and revised European guidelines do not include size as a basis of their surveillance interval recommendations [57,58] (Table 4). There is limited evidence to support the recommended surveillance intervals in the guidelines. It is likely that this schedule is overly intensive with associated healthcare costs for some patients. For others, this schedule may not be intensive enough and they may develop an interval cancer. All patients should be made aware when entering surveillance programs, that in rare cases, a cancer could develop between surveillance imaging. They should contact their medical team prior to their next imaging study if they develop any new symptoms in the interim period.

Recent studies on MCNs have shown that the risk of cancer in cysts less than 40 mm in size and without worrisome features is exceedingly rare [84,118]; therefore, in contrast to other guidelines, the revised European guidelines recommends surveillance of all MCNs <40 mm, following the same surveillance intervals as for a BD IPMN [57].

### 3.2. When Can Surveillance Be Stopped?

Although the potential of IPMNs to progress to invasive cancer is clearly recognized, there remains controversy over which guidelines should be followed. The AGA guidelines recommend for the discontinuation of surveillance at 5 years in cysts less 3 cm in the absence of MPD duct dilation and mural nodule. This recommendation was in contrast to the other guidelines [58] and based on a single study of patients with less than 2 cm cysts without worrisome features, in whom none were found to develop invasive cancer after a surveillance period beyond 5 years [119] [Table 4]. Recent studies have varied. Some larger studies of patients with well-characterized IPMNs have disputed this finding by demonstrating a risk of malignant transformation that persists beyond 5 years and which probably increases over time [18,19,22,120,121,122,123]. The largest retrospective study to date of 1404 patients with a clinically defined IPMN found an incidence of malignant transformation of 2.9%, 5.9% and 14% at 5, 10 and 15 years, respectively [39]. As part of secondary analyses, the authors also demonstrated that patients with low risk BD-IPMN <15 mm, had a cumulative incidence rates of pancreatic carcinoma 2.2%, 4.6%, and 7.4% at 5, 10, and 15 years, respectively [39]. In contrast, a recent multicenter study of 806 patients with BD-IPMN ≤15 mm at diagnosis who do not develop worrisome features had an overall risk of malignancy of 1.7% over a 5 year median follow-up, with a cumulative incidence of malignancy of 0.94% at 5 years and 3.37% at 10 years [124]. This is similar to other studies that have suggested cysts can be risk-stratified based on size [22,125,126]. Regardless, the International Consensus and European guidelines recommend continued surveillance in all patients with an IPMN/MCN, as long as they are fit to undergo surgical resection [30,57].

### 3.3. Surgical Resection in IPMN/MCN

#### 3.3.1. Indications for Surgical Resection

The indications for surgery for patients with IPMN or MCN differ between guidelines but absolute and relative indications for surgery are summarized in Table 3. The International Consensus guidelines define “high risk features” as obstructive jaundice, MPD as greater than 10 mm, positive cytology or an enhancing mural nodule ≥5 mm. If any of these high-risk features are present, they advocate direct surgical referral without further testing. An EUS is advised if any “worrisome features” are present, which includes; cyst growth rate ≥5 mm over 2 years, increased levels of serum CA19-9, MPD dilation between 5 and 9 mm, cyst diameter ≥30 mm, acute pancreatitis (attributable to the IPMN), enhancing mural nodule of <5 mm, an abrupt change in diameter of MPD with distal atrophy, lymphadenopathy or thickened or enhancing cyst walls [30]. The European guidelines, published in 2018, define absolute indications for surgery as positive cytology for malignancy, the presence of a solid mass, obstructive jaundice, an enhancing mural nodule (≥5 mm) or MPD dilatation ≥10 mm. Relative indications for surgery include a growth rate ≥5 mm per year, elevated serum CA19-9 (>37 U/mL), MPD dilatation between 5 and 9.9 mm, cyst diameter ≥40 mm, new-onset diabetes mellitus, acute pancreatitis caused by IPMN or an enhancing mural nodule (<5 mm). If patients have no comorbidity, a lower threshold for surgery is advocated of just one relative indication. In patients with significant comorbidity, more than one relative indication is required to proceed to surgery and if only one relative indication was present, then close surveillance is advised with CA19-9 and MRI with or without an endoscopic ultrasound (EUS) examination [57].

The ACR guidelines define absolute indications for surgery as obstructive jaundice, dilated MPD, positive cytology showing cancer, an enhancing nodule or solid mass. Relative indications included an elevated CA 19-9, new-onset diabetes, acute pancreatitis, cyst growth >5 mm per 2 years, MPD 5–9 mm, cyst >4 cm, or enhancing nodule <5 mm [60]. ACG has set recommendation for referral to surgery which includes obstructive jaundice, acute pancreatitis, solid mass, MPD >5 mm, cyst >3 cm, change in MPD with upstream atrophy, positive cytology showing cancer and presence of a mural nodule but strongly advocates patients are discussed in a multidisciplinary setting prior to surgery [59]. Lastly, the AGA guidelines recommends consideration of surgery if there are two or more of the following features present: dilated MPD, cyst >3 cm, and/or mural nodule. Unlike the other guidelines, the AGA are more conservative and do not recommend resection for MPD dilatation alone, and require the presence of a mural nodule or positive cytology as well [58]. Many of the differences in the recommendations between the guidelines arise because they are based on low or very low quality evidence due to a lack of well characterized prospectively followed cohorts of patients with PCL.

#### 3.3.2. Surgery

High-risk cystic lesions in the head or in the uncinate process of the pancreas typically undergo a pancreatoduodenectomy, whereas a distal pancreatectomy with splenectomy is performed for cysts located in the body or tail of the pancreas. A conventional pancreatoduodenectomy involves removing the pancreatic head, duodenum, part of the jejunum, common bile duct, gallbladder as well as performing a partial gastrectomy, and can be performed open or by minimally invasive laparoscopic or robotic approaches. A distal pancreatectomy involves the removal of the body and tail of the pancreas to the left of the superior mesenteric artery and vein and can also be accomplished using open or minimally invasive approaches. Surgical resection of an IPMN or MCN is associated with a perioperative morbidity of 20–40% and mortality of 1–3% for pancreatoduodenectomy and <1% for distal pancreatectomy [127,128] in high volume centers. Less extensive resections, such as a central pancreatectomy or enucleation, can be performed as a parenchyma-sparing technique. This is a potentially attractive approach because of the potential for improved post-operative pancreatic function. Unfortunately, post-operative morbidity and mortality is similar or higher due to the significant risk of pancreatic fistula. Therefore, this procedure is only performed in select young patients [129].

Although IPMNs can extend along the MPD or be a multifocal disease, none of the current guidelines currently recommend a total pancreatectomy due to the morbidity associated with patient being rendered diabetic and having definite postoperative endocrine insufficiency [30,57]. Despite recommendations, in an international expert survey, around half of the respondents suggested that in certain situations, they would advise total pancreatectomy [130], mainly for IPMN with MPD involvement in order to reduce the risk of recurrence. Indications for surgery in BD-IPMN also differ between the guidelines and are summarized in Table 4.

The International Consensus and AGA guidelines recommend resection of MCN regardless of size whereas the revised European guidelines support surveillance of MCN <40 mm without concerning features, following the same surveillance intervals as for a BD IPMN [30,58].

Several studies have evaluated the accuracy of the different guidelines at predicting advanced neoplasia based on the recommended indications for surgery in IPMN [131,132,133]. These studies recognized that all current guidelines lead to surgical overtreatment of IPMNs. In a comparative study, the AGA guidelines appears to have a significant risk of missing patients with advanced neoplasia (12–45%), although fewer patients would have undergone unnecessary surgery [131,132]. Our center, like many other large hepatopancreaticobiliary centers, broadly follows the International Consensus and European guidelines and discusses each patient with high-risk or worrisome features at a regular multidisciplinary meeting prior to surgical resection [134].

#### 3.3.3. Follow Up after Surgery and Predictors of Recurrence

IPMNs without invasive cancer, recur even after surgery in contrast to MCNs, which do not recur. In a study of 130 patients followed for a median of 38 months, 17% developed imaging evidence of a new or progressive IPMN. Eight percent ultimately underwent a completion pancreatectomy and of those 27% (3 patients) had invasive cancer. In addition, two further patients developed metastatic pancreatic adenocarcinoma and did not undergo resection. All 5 patients (4%) that developed cancer had negative margins after the initial operation. The presence of a negative margin did not significantly affect whether patients developed a recurrence of IPMN. A family history of pancreatic cancer was predictive of developing a new IPMN (23% vs. 7% (*p* < 0.05)). The chances of developing a new IPMN at 1, 5, and 10 years after the initial surgery was 4%, 25%, and 62%, respectively, and the estimated chances of developing invasive cancer at 1, 5, and 10 years after surgery was 0%, 7%, and 38%, respectively [135]. In a multicenter study of 126 patients undergoing resection for a non-invasive IPMN, followed for a median of 9.5 years, a family history of pancreatic cancer (hazard ratio 3.05) and high-grade IPMN (hazard ratio 1.88) were risk factors for recurrence. Again, a positive margin alone was not predictive of recurrence, but the extent and grade of dysplasia at the margin did significantly predict recurrence. Of note, 74% of recurrences occurred after 3 years and 32% after 5 years, supporting long-term surveillance post resection [136].

The European guidelines advocate that patients undergoing a resection of a BD-IPMN with LGD or Intermediate grade dysplasia (IGD) be followed in the same way, as an unresected IPMN and surveillance should be continued as long as patients are fit to undergo a completion pancreatectomy. For patients with HGD, MT IPMN or MD IPMN, follow-up with cross-sectional imaging of the remnant pancreas every 6 months for the first 2 years is recommended, followed by yearly surveillance after that, as long as imaging findings are stable [57]. The International Consensus guidelines advocate enhanced follow- up with at least twice a year imaging in patients with a family history of pancreatic cancer, a surgical resection margin with HGD, and a resected IPMN of a no intestinal subtype. A follow-up every 6 to 12 months in all other patients with resected IPMNs is recommended [30] An IPMN-associated cancer should be followed up in the same way as follow-up for PDAC after pancreatectomy [57] which involves undertaking cross sectional imaging and measuring CA19-9 every 3–4 months [137].

A systematic review of 13 studies with 773 patients with a MCN, found no risk of recurrence after resection of MCN without pancreatic cancer [29]. Thus, patients with surgically resected MCN whether with LGD or HGD, do not require surveillance. Patients with invasive cancer arising from a surgically resected MCN have a 25% risk of cancer recurrence [84]. Therefore, the International Consensus and European guidelines advocate patients with an MCN- associated cancer should be followed in the same way as those with an IPMN-associated cancer or pancreatic cancer after a partial pancreatectomy [30,57].

### 3.4. Cyst Ablation

While surgery is the only curative treatment for resectable high-risk IPMNs or MCNs, it is associated with significant morbidity and mortality. In addition, despite current recommendations, approximately 25% of patients who have a presumed IPMN or MCN surgically resected are ultimately found to have a cyst with no malignant potential [138]. This is even more prevalent in resected BD IPMN where up to 78% do not have high-grade dysplasia or invasive cancer [139]. There has therefore been a growing interest in minimally invasive alternative approaches, especially for those patients who are elderly or have high-risk surgical candidates [140,141]. One such method is cystic ablation of the PCL.

Cystic ablation has primarily been undertaken by injecting alcohol or a chemotherapy agent directly into the cyst under EUS-guidance, with the aim of disrupting the epithelial cyst lining leading to cyst resolution [142,143,144,145,146,147,148,149,150]. A recent meta-analysis found the pooled rate of complete resolution of cysts treated with alcohol and paclitaxel was 63.6%, but dropped to 32.8% if ethanol alone was used. Adverse events were reported in approximately 15%, mostly due to acute pancreatitis [151]. An important limitation of many of the single center studies is that follow up was relatively short [142,143,144,145,146,147,148,149,150], but longer-term data is emerging. Choi and colleagues have reported the largest series to date of 164 patients. Patients were followed up for 71 months and complete radiological ablation was achieved in 99% [152]. The highest rate of successful ablation has been in MCNs [151]. This is likely because there are more septations in serous cystic neoplasms and IPMNs, which stop the fluid diffusing through the whole cyst and limiting the ablation of the epithelial lining. Radiofrequency ablation (RFA) causes tissue destruction by the application of a high frequency alternating current, which generates heat leading to a coagulative necrosis. RFA catheters that can be passed through the working channel of a linear echoendoscope have been developed to enable targeted RFA under EUS guidance. Two small prospective studies, with follow up of less than 12 months, evaluated the safety and efficacy of this technique in the management of PCL. Complete ablation was reported between 33–65%, with adverse events occurring in 0–10% [153,154].

Although these initial studies are promising, there continues to be a concern about partial treatment when ablating PCLs. In addition, changes in cyst shape after treatment make it challenging to define complete ablation radiographically. At present, these treatments are only recommended for use within research protocols and formal registries [155]. Further studies are required to determine the long-term efficacy and the clinical benefits of these treatments as well as their place in management protocols [155].

## 4. Conclusions

The detection and surveillance of IPMNs and MCNs enables the early detection of high-risk lesions and potentially curative surgical resection in a disease with a dismal prognosis. However, differentiating premalignant and malignant cysts from non-malignant cysts continues to be challenging but additional diagnostic tools are emerging. There is no single algorithm for the management of these lesions, largely due to the lack of high quality of evidence on which to base recommendations. The current guidelines for the management of IPMNs and MCNs differ in their recommendations for surgical resection, surveillance protocols and if or when to stop surveillance. Ongoing prospective studies will be able to modify these recommendations and future guidelines. Clinicians should therefore have a low threshold for discussing these patients with a multidisciplinary team who have expertise in diagnosis and surgical treatment of these PCL, especially when worrisome features are present. Minimally invasive surgical approaches and novel endoscopic ablative treatments show promise, but their utility requires further validation in larger studies.

## Figures and Tables

**Table 1 jcm-10-01284-t001:** Key clinical and imaging features of common pancreatic cystic lesions.

	Intraductal Papillary Mucinous Neoplasm (IPMN)	Mucinous Cystic Neoplasm (MCN)	Serous Cystic Adenoma	Pseudocyst	Cystic Pancreatic Neuroendocrine Tumor	Solid Pseudopapillary Neoplasm
**Sex**	M or F	F	F	M or F	M or F	F
**Age**	65	40	60	-	50	30
**Pancreatic localization**	Head	Body/Tail	Throughout	Throughout	Throughout	Body/Tail
**Typical imaging features**	MD-IPMN: Dilated MPD SB IPMN: Dilated side branch or cyst that connects to the MPD	Unilocular, macrocystic	Microcystic (honeycomb) appearance	Unilocular cyst, sometimes with necrotic debris	Solid cystic lesion, hypervascular	Solid cystic lesion
**Communication with the MPD**	+	−	−	+ or −	−	−
**Solitary/multifocal**	Solitary/multifocal	Solitary	Solitary	Solitary	Solitary	Solitary
**Malignant potential (surgically resected lesions) ***	MD/MT IPMN: 36–100% BD IPMN: 11–30%	10–39%	0%	0%	10%	10–15%

* [29,30,31,32]; BD-IPMN: branch duct IPMN; MPD: main pancreatic duct; MD-IPMN: Main duct IPMN; MT-IPMN: Mixed type IPMN.

**Table 2 jcm-10-01284-t002:** Pathological subtypes of IPMN.

Subtype	Papillae	Mimicker	Typical Level of Atypia	MUC Staining
Gastric	Thick fingerlike or small tubules	Foveolar gland or pyloric gland	LGD	MUC5AC MUC6
Intestinal	Villous	Intestinal villous neoplasm	IGD / HGD	MUC2 MUC5AC
Pancreaticobiliary	Fern like	Cholangiopapillary neoplasm	HGD	MUC1 MUC5AC MUC6
Oncocytic	Pylloid	Oncocytic tumor	HGD	MUC5AC MUC6 (+/− MUC1 or MUC2)

HGD: high grade dysplasia, IGD: intermediate grade dysplasia; LGD: low grade dysplasia.

**Table 3 jcm-10-01284-t003:** Indications for surgical resection in IPMN or MCN as outlined by current guidelines.

Guideline	Cyst Type	Absolute Indications for Surgery	Relative Indications for Surgery
American Gastroenterology Association (2015) [58]	MCN	All MCN	-
	IPMN	MPD ≥5 mm (on MRI and EUS) and solid component Cytology positive for malignancy	-
International Consensus Guidelines (2017) [30]	MCN	All MCN	-
	IPMN	Cytology suspicious or positive for malignancy Jaundice (tumor-related) Enhancing mural nodule (≥5 mm) MPD dilatation ≥10 mm	Growth rate ≥5 mm over 2 years Increased levels of serum CA19-9 PD dilatation between 5 and 9 mm Cyst diameter ≥30 mm Acute pancreatitis (caused by IPMN) Enhancing mural nodule (<5 mm) Abrupt change in diameter of MPD with distal atrophy Lymphadenopathy Thickened or enhancing cyst walls
European (2018) [57]	MCN	Cyst diameter ≥40 mm Enhancing mural nodule Symptoms (jaundice, acute pancreatitis, new- onset diabetes mellitus)	
	IPMN	Positive cytology for malignancy or HGD Solid mass Jaundice (tumor- related) Enhancing mural nodule (≥5 mm) MPD dilatation ≥10 mm	Growth rate ≥5 mm per year Increased levels of serum CA19-9 (>37 U/mL) MPD dilatation between 5 and 9.9 mm Cyst diameter ≥40 mm New- onset diabetes mellitus Acute pancreatitis (caused by IPMN) Enhancing mural nodule (<5 mm)
American College Gastroenterology (2018) [59]	IPMN or MCN	-	Indication for multidisciplinary review: Jaundice secondary to the cyst Acute pancreatitis secondary to the cyst Significantly elevated serum CA19-9 Any of the following imaging findings: mural nodule, solid component, dilation of MPD >5 mm, focal dilation of the MPD, mucin-producing cysts ≥3 cm. The presence of HGD or pancreatic cancer on cytology
Radiology White paper (2017) [60]	IPMN or MCN	Jaundice Enhancing mural nodule MPD >10 mm	Indications for EUS-FNA: High risk features: mural nodules, wall thickening, MPD >7 mm, peripheral calcium, cyst >2.5 cm Interval growth (>20% in longitudinal axis)

IPMN: Intraductal papillary mucinous neoplasm; MCN: Mucinous cystic neoplasm; MPD: main pancreatic duct; EUS-FNA: Endoscopic ultrasound and fine needle aspiration; HGD: High Grade Dysplasia; MRI: Magnetic Resonance Imaging.

**Table 4 jcm-10-01284-t004:** Comparison of the guideline recommendations for surveillance protocols and indications for EUS.

Guideline	Surveillance Protocol	Indication for EUS	Discharge from Surveillance
**American Gastroenterology Association (2015) [58]**	Patients with pancreatic cysts <3 cm without a solid component or a dilated pancreatic duct should undergo MRI in 1 year, then every 2 years, for a total of 5 years if there is no change in size or characteristics.	Pancreatic cysts with at least 2 high-risk features, such as size >3 cm, a dilated (or increasingly dilated) main pancreatic duct, or the presence of an associated solid component	Discharge if there has been no significant change in the characteristics of the cyst after 5 years of surveillance or if the patient is no longer a surgical candidate
**International Consensus Guidelines** **(2017) [30]**	In cysts without worrisome features: <1 cm: CT / MRI in 6 months, then every 2 years if no change 1–2 cm: CT / MRI 6 monthly for 1 year, yearly for 2 years, then every 2 years if no change 2–3 cm: EUS in 3-6 months, then in 1 year if no change, alternating MRI with EUS. Consider surgery in young, fit patients with need for prolonged surveillance. >3 cm: Alternating MRI with EUS every 3–6 months. Strongly consider surgery in young, fit patients	If one or more of the following “worrisome features” are present: Acute Pancreatitis Cyst >3 cm∙ Enhancing mural nodule <5 mm Thickened/enhancing cyst walls Main duct size 5–9 mm Abrupt change in caliber of pancreatic duct with distal pancreatic atrophy Lymphadenopathy Increased serum level of CA19-9 Cyst growth rate > 5 mm/2 years	Continue as long as patients are fit to undergo surgical resection
**European** **(2018) [57]**	1st year after diagnosis: Clinical evaluation, serum CA19-9, MRI or EUS every 6 months. After 1 year + no indications for surgery: Clinical evaluation, serum CA19-9 and MRI or EUS annually	EUS-FNA should only be performed when the results are expected to change clinical management. EUS-FNA should not be performed if the diagnosis is already established by cross-sectional imaging, or where there is a clear indication for surgery	Continue as long as patients are fit to undergo surgical resection
**American College Gastroenterology (2018) [59]**	In patients with a presumed IPMN/MCN without concerning features or indications for surgery: <1 cm MRI in 2 years 1–2cm MRI in 1 year 2–3 cm MRI or EUS in 6–12 months	EUS-FNA can be considered if the diagnosis is unclear, and results will alter management. Cyst fluid CEA can differentiate IPMN/MCN from other cysts. Cytology can assess for the presence of HGD or pancreatic cancer. Molecular markers can help identify IPMNs / MCNs in cases where it will change management	Continue as long as patients are fit to undergo surgical resection
**Radiology White** **paper (2017) [60]**	Pancreatic cyst without features of concern: <2 cm imaging every 1–2 years depending on age and length of size stability >2 cm imaging every 6 months for 2 years, then annually for 2 years then every 2 years.	Increasing cyst size, the presence of “worrisome features” or “high-risk stigmata,” should prompt EUS FNA	Continue as long as patients are fit to undergo surgical resection. Stop surveillance if cyst <1.5 cm and stable over 10 years of surveillance

IPMN: Intraductal papillary mucinous neoplasm; MCN: Mucinous cystic neoplasm; MPD: main pancreatic duct; EUS-FNA: Endoscopic ultrasound and fine needle aspiration; HGD: High Grade Dysplasia; MRI: Magnetic Resonance Imaging.

## Data Availability

Not applicable.

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
