# Peer review of "A Review of the Diagnosis and Management of Premalignant Pancreatic Cystic Lesions"

_jcm, 2021, doi:10.3390/jcm10061284_

Round 1
Reviewer 1 Report
The authors present a detailed and comprehensive review of premalignant pancreatic cysts. The content is up to date and well referenced.
I have 2 major concerns.
1. The complex nature of the subject and conflict in guidance has led to the length of the article to be excessive and makes it difficult to read or be used as an educational resource. I would suggest reducing the length by at least 1000-2000 words.
2. Given the length of the article and the conflict of guidelines detailed, as a general clinician accessing the article I would have trouble knowing how to manage a cyst at the end of reading it. Although there are differences in guideline recorded in the article there also are a number of themes and key points that would be beneficial to be summarized - possibly in a table.
Reviewer 2 Report
The authors reviewed the diagnosis and management of premalignant pancreatic cystic lesions by comparing major guidelines and evidence. The review is well written and well balanced including latest endoscopic technologies such as needle-based confocal laser endomicroscopy and through the needle microbiopsy forceps. However, I think there are a few improvements that should be made.
・The authors compared the uncertainty between the guidelines including indications for surgery, surveillance protocols and if and when surveillance can be discontinued. There is a table comparing the indications for surgery in each guideline but differences in surveillance protocols and if and when surveillance can be discontinued should also be compared in a table as well as it would be easier for readers to understand at a glance. ・There are a few typographical errors that should be corrected. For example, CA19-9 is written "Ca19-9" in L494 of P12 and Conclusion is misspelled "Conculusion" in L623 of P14.
Reviewer 3 Report
Keane and Afghani performed an interesting and complete review about the diagnosis of premalignant pancreatic cystic lesions (PCLs). This review ranges in various topics regarding PCLs. Anyway, in my opinion, some issues should be further addressed.
- Paragraph 1.2: expand this section specifying where specific mutations can be found (bloodstream, cystic fluid, cystic walls,…) and which can be the clinical implication of these mutations
- In the section about different available guidelines, try to expose the peculiar differences in terms of diagnostic capacity of each one (I.e., which is the one with the highest possibility of false-positive cases, which is the most surgical aggressive one) with the connected risks.
- Paragraph 2.1.3: expand this section speaking about the role of abrupt change in main pancreatic duct caliber, other than its only dilation
- In the same section deserve a sub-paragraph on how to differentiate IPMN-MD/mixed-type from chronic pancreatitis with MPD dilation.
- Expand the paragraph on the role of EUS focusing also on the different needles that can be used to obtain a diagnosis such as FNA, FNB, brushing,… (doi: 10.4103/eus.eus_45_18 ; PMID: 30323157)
- In paragraph 3.1, you should underline, according to multiple papers available in the literature, the better diagnostic capacity with fewer risks of EUS with respect to CT-scan.
- Underline and quantify the risk of interval cancers during the surveillance
- Try to write a small paragraph about PCL management's possible future scenario (artificial intelligence, NGS,…)
